# Psychological and Physiological Signatures of Music Listening in Different Listening Environments—An Exploratory Study

**DOI:** 10.3390/brainsci11050593

**Published:** 2021-05-03

**Authors:** Mari Tervaniemi, Tommi Makkonen, Peixin Nie

**Affiliations:** 1Faculty of Educational Sciences, University of Helsinki, P.O. Box 9, 00014 Helsinki, Finland; peixin.nie@helsinki.fi; 2Cognitive Brain Research Unit, Department of Psychology and Logopedics, Faculty of Medicine, University of Helsinki, P.O. Box 21, 00014 Helsinki, Finland; tommi.makkonen@helsinki.fi

**Keywords:** music, emotion, emotion regulation, stress reduction, cortisol

## Abstract

We compared music emotion ratings and their physiological correlates when the participants listened to music at home and in the laboratory. We hypothesized that music emotions are stronger in a familiar environment, that is, at home. Participants listened to their self-selected favorite and neutral music excerpts at home and in the laboratory for 10 min in each environment. They completed the questionnaires about their emotional states and gave saliva samples for the analyses of the stress hormone cortisol. We found that in the context of music listening, the participants’ emotion ratings differed between home and the laboratory. Furthermore, the cortisol levels were generally lower at home than in the laboratory and decreased after music listening at home and in the laboratory. However, the modulatory effects of music listening on cortisol levels did not differ between the home and the laboratory. Our exploratory multimethodological data offer novel insight about the psychological and physiological consequences of music listening. These data reveal the sensitivity of the current research methods to investigate human emotions in various contexts without excluding the use of laboratory environment in investigating them.

## 1. Introduction

Thanks to mobile technology, we have access to music listening everyday everywhere 24/7. Despite the rapid increase of the use of these online streaming services during the past decade, music recordings still have huge market; e.g., an estimation for the global trade revenue of recorded music in 2019 was US$20.2 billion [1]. In addition, concerts and other live music events such as opera and musicals attract large audiences to enjoy national and international performances. Obviously, we are deeply attracted by music, but why?

One motivation for music listening in our daily life might result from the power music has in modulating our mental state in terms of emotions and vigilance. Consequently, in music psychology, music listening has been conceptualized and empirically investigated in at least three frameworks, namely, music emotions, mood regulation, and stress reduction. These frameworks will be briefly illustrated below (for a broader review, see [2,3]).

First, music emotion literature is focused on investigating which emotions can be associated with a given music excerpt by its listeners. Questionnaire measures used while music listening and also retrospectively showed that music listening is linked to basic emotions (e.g., joy), specific feelings (nostalgia), changes in the vigilance level (“soothing”), as well as ratings in terms of valence (“felt good”) [4,5]. In many physiological and brain mapping studies, the ultimate pleasure caused by music has been operationalized by the occurrence of chills which are experienced by some individuals while listening to their favorite excerpts [6,7,8,9].

Second, in the mood regulation literature, questionnaire and interview data showed that people use music to modify their emotional state intuitively and without specific instruction, e.g., to soothe, to comfort, or to cheer up—all of this along the life span from adolescence until late adulthood [10,11,12,13].

Third, and most importantly in the current context, in music medicine, by using questionnaires and physiological measures such as cortisol analyses, music listening has resulted in a noticeable stress reduction as indicated by psychological self-reports and concomitant biological stress measures [14]. In this context, there is also some evidence that, particularly, favorite music might help to reduce pain and anxiousness more than any other music [15]. However, musical preference might not be the same in all listening situations, especially when attended external stressors are included [16]. Moreover, the emotional effects of music listening are also largely dependent on the types of music as well as experimental settings (see e.g., [17,18,19,20,21]). Thus, in our view point, studies on the emotional effects of music listening taking into account the music preferences and different listening situations are in the need to be promoted further.

Traditionally, listening and physiological studies have been conducted in the laboratory environment. However, one might ask whether emotional reactions of listeners are comparable in a neutral unfamiliar environment and in an environment in which listening usually takes place. Recently Azhari and others showed higher metabolic brain activity in the prefrontal cortex (involved in processing of contextual information) if emotional infant and adult vocalizations were played in outdoor than in domestic contexts [22]. This finding corroborates and extends their findings indicating that physiological signals to such emotional cues are also stronger in outdoor than in domestic environments [23].

Regarding music listening, the first pioneering empirical endeavors to obtain information about listeners’ emotional responses and to record an electroencephalogram (EEG) in a concert setting were accomplished by Dolan and others [24]. They found that the complexity of the EEG signal, reflecting alertness, was higher when the music excerpts included improvisational elements than when they were played as denoted by the musical score. Ratings of the music performance reflected a preference for improvised performance. These studies also had an intent to investigate music induced emotions in an ecologically valid manner, yet, without an attempt to compare the music emotions in different listening environments.

Consequently, from methodological perspective, it is currently challenging to get a unified perspective about the affordances of music in modulating our physiological and psychological state. While questionnaire and interview studies are conducted in laboratory settings as well as in regular listening environments, physiological studies including those with hormonal analyses have been conducted mainly in the laboratory (see, however the recent report about an ambulatory, home-performed, cortisol study by Wuttke-Linnemann and others [25]). Thus, one might speculate about the differential effects the listening environment has on the results obtained—it might be that the modulatory effects of music on its listener are stronger in a familiar environment, that is, at home, when compared with the laboratory [26].

Here, our primary interest was to investigate whether music listening modulates music emotion ratings differentially in two different listening environments, namely, at home vs. in the laboratory. In addition to this questionnaire-based part of the study, we also compared the extent to which favorite vs. neutral music listening modifies the salivary cortisol levels of the listeners at home and in the laboratory. To this end, our specific instruction for participants was that one of the music excerpts should be their favorite music, expected to evoke more positive emotions than the other music excerpt, instructed to be neutral and not to evoke any specific emotions.

## 2. Materials and Methods

### 2.1. Participants

Participants were adult healthy volunteers who all had some experience in music making (e.g., as part of their hobbies or as part of their elementary school curriculum) but were not currently actively practicing or performing music. They were recruited by using e-mail lists of local universities and other academic organizations as well as by direct contacts by the research assistants SG and HM. They were selected on the basis of their availability and interest on participation, the participants thus forming a convenience sample. In total, there were 37 participants who were 20–40 years of age (mean 26.4 years, SD 4.4 years; 19 male).

Before the commencement of the project, the study protocol was reviewed and approved by the ethics committee of the former Faculty of Behavioural Sciences, University of Helsinki (01/2011). Informed written consent was obtained from all participants and all research was performed in accordance with relevant guidelines.

### 2.2. Experimental Procedure

Before coming to the laboratory, the participants were provided with four saliva sample tubes (Salivette) together with the instructions of giving the samples the night before (one sample) and in the morning (3 samples) of the laboratory visit. These samples were used for the pilot analyses of other hormonal values, not reported here. The participants also received the questionnaires for background information and the daily use of music. They were asked to bring both questionnaires and the tubes to the laboratory. Additionally, they were asked to bring two excerpts of music to the laboratory by using a CD or as wav files; one of these excerpts representing their favorite music and the other one as neutral as possible.

The laboratory investigation was always conducted between noon and 4 pm. The duration of the whole experiment was 105 min on average.

First the questionnaires (see below) and sound files were collected. If the files were in mp3 format, they were converted to wav by using Adobe Audition 2 (Adobe Inc, San José, CA, USA). The intensity of the audio output was manually calibrated to be approximately 60 dB. Audiometry was conducted using Oscilla SM950 (Natus Medical Denmark APS, Taastrup, Denmark) (15 participants) and Oscilla USB350SP (23 participants). In all these participants, their hearing was normal (<25 dB, 125 Hz–8 kHz). For two participants the audiometry was not conducted due to unavailability of the device. However, these participants reported their hearing to be normal.

Electrocardiogram (ECG; e.g., see [27]) and electrodermal activity (EDA) were measured using BioSemi ActiveTwo system; these data will be reported in a separate paper.

After ECG and EDA electrodes were attached, music excerpts copied to the PC, and background questionnaires collected, the participant was seated (reclined position in an armchair) in an acoustically and electronically shielded chamber. S/he was instructed to have a rest of 10 min and to glance the magazines available. Baseline data for ECD and EDA were recorded for 5 min.

After this 10-min rest, the participant gave baseline saliva samples that were kept in dry ice and was asked to fill a questionnaire about the current emotional state. Note that they spent at least 20 min at the laboratory before the first saliva collection thus minimizing the effects of physical activity on their cortisol levels. They were also instructed to not eat, drink, or smoke during one hour before entering the laboratory in order to optimize the saliva sample collection.

Music listening (favorite music; neutral music) were given in random order. Both excerpts were played on repeat for 10 min on a headphone (Sony MDR-7506).

After the listening of the first excerpt, the participant was asked to fill a questionnaire about the current emotional state. Baseline data were recorded again for 5 min. After this 20-min rest, the participant gave the saliva sample.

Similarly resting data were recorded after the second excerpt (ECG and EDA data acquisition 5 min, total rest duration 20 min). After that, they were requested to fill in the questionnaire about their emotional state and to give the last saliva sample.

After the laboratory experiment was completed, instructions were given for the next identical saliva sample collection to be performed at home before and after music listening within 2–7 days. This part of the study was instructed to take place between noon and 4 pm.

As in the laboratory, also at home they were requested to first give the baseline saliva sample, fill in the emotional state questionnaire, rest, and start listening to the first excerpt (randomized to be either favorite or neutral) for 10 min. Music listening was instructed to be uninterrupted and focused, however, everyday background sounds were not a problem. After giving the saliva sample, they were asked to wait for 20 min, fill in the questionnaire and start listening to the second excerpt for 10 min. After this, they were asked to fill in the questionnaire again and to give the last saliva sample. These data were successfully collected from 28 out of 37 participants.

In the questionnaire on emotion ratings, the participants were asked to rate their current emotional state along the following attributes along a scale from 1 (least) to 5 (most): excited, peaceful, enthusiastic, happy, sad, melancholic, irritated, tired, satisfied, curious, emotionally moved, and sentimental. These attributes were carefully selected from wider sets of attributes used in other emotional state inventories. In the statistical analyses, we grouped part of them to high-arousal (happy, excited, enthusiastic, irritated) and low-arousal (sad, melancholic, peaceful, satisfied) emotion scores as well as to positive valence (excited, peaceful, enthusiastic, and happy) and negative valence (sad, melancholic, irritated, tired) scores.

### 2.3. Data Analyses

Saliva samples were delivered to the laboratory of the Finnish Institute of Occupational Health (Helsinki, Finland) in a freezer bag filled with dry ice on a daily basis. Salivary cortisol was analyzed there with chemiluminescence immunoassay (LIA, IBL Hamburg, Germany) based on the competition principle. An unknown amount of antigen present in the sample and fixed amount of enzyme labelled antigen compete for the binding sites of the antibodies coated onto the wells. Measuring range of the method was 0.43 to 110 nmol/L. The coefficient of variation % of intra- and inter-assay of the method was 5 and 8%, respectively.

Normality of the cortisol sample distributions was tested with the Kolmogorov-Smirnov test. As several of the variables were not normally distributed, a logarithmic transformation was made all variables used in the analysis. After that, the distributions were found to be normal. To determine the putative effect of the order of listening to favorite vs. neutral music, we first conducted 2 (favorite, neutral) × 2 (order 1st, 2nd) ANOVA for the cortisol values. It indicated no interaction between the order and type of the music in the laboratory (F(1,35) = 0.026, *p* = 0.873) or at home (F(1,27) = 0.042, *p* = 0.838). Therefore, we report the statistical analyses which have been performed for the data after pooling the cortisol data across the listening order.

Normality of the emotion ranking scores was tested with the Kolmogorov-Smirnov test. As several of them were not normally distributed, we used non-parametric Wilcoxon signed-rank tests separately for high- and low-arousal emotion scores as well as for positive- and negative-valence emotions scores. These analyses were separately conducted for baseline ratings (before listening) and after listening to the favorite and neutral music excerpts between the listening at home and in the laboratory.

## 3. Results

### 3.1. Effects of Music Listening on Emotion Ratings

The mean values of emotion scores differed statistically significantly between different listening environments particularly at the baseline stage when low arousal and positive valence emotion scores were higher in the laboratory than at home and negative valence emotion scores were lower in the laboratory than at home (Table 1). Additionally the high-arousal emotion score significantly differed between laboratory and home after listening to neutral music, the score being higher at home than in the laboratory. Marginally different emotion scores in different listening environments were obtained for positive- and negative-valence scores after listening to the favorite music: positive valence scores were marginally higher in the laboratory than at home and negative valence emotion scores were marginally lower in the laboratory than at home.

### 3.2. Effects of Music Listening on Cortisol Levels

As depicted in Figure 1, music listening reduced the cortisol levels significantly. This was indicated by the main effect of condition in the cortisol levels when baseline and post-listening values were compared (2-way repeated measures ANOVA with factors condition (baseline, favorite music, neutral music) × listening environment (home, laboratory); (F(2,56) = 24.535, *p* < 0.001, η^2^ = 0.467). In the post hoc tests, the cortisol levels after music listening were significantly lower than at the baseline (*p* < 0.05 between baseline and neutral music as well as between baseline and favorite music), however, without a significant difference between the favorite and neutral music listening (*p* = 0.535).

The music listening environment also significantly modulated cortisol levels which were lower at home than in the laboratory (main effect of the listening environment (F(1,28) = 5.190, *p* < 0.05, η^2^ = 0.156). In post-hoc tests, this difference was observed at the baseline as well as after listening to the neutral and favorite music (*p* < 0.05 in all comparisons).

Against our expectations, there was no interaction between listening environment and music listening in the decrease of cortisol levels caused by music listening, implying that the effect of music to reduce cortisol levels did not differ between home and the laboratory (F(2,56) = 0.222, *p* = 0.802, η^2^ = 0.008).

## 4. Discussion

Our aim was to investigate whether music listening modulates emotional state ratings and cortisol levels differentially in two different listening environments. We expected that the effects of music listening on emotional ratings and on cortisol levels would be more profound at home than in the laboratory. However, our results show that while music listening modulated the cortisol levels, there was no significant difference in this modulation as observed at home or in the laboratory. Furthermore, our results show that the baseline emotion ratings differed between home and the laboratory and that the emotion ratings after listening to the favorite music excerpts tended to differ between home and the laboratory.

Thus, the preference for a given music might be considered an important factor for psychological and physiological concomitants of the emotional reactions in its listener. Supporting findings about stronger emotional reactions while listening to one’s favorite music were previously given in the musical chill paradigm by the work of Zatorre group. They showed that the occurrence of chills while listening to self-selected music was paralleled with elevated reactions of the autonomous nervous system and reward-related brain processes [8]. Since then, the key role of dopaminergic processes as the basis of pleasure caused by music has been determined [9,28]. In parallel, Wilkins and others showed that the default mode network (especially precuneus) was more connected when listening to preferred music than when listening to music with less emotional meaning [29]. The health benefits of favorite music listening were originally shown by Särkämö and others who reported faster emotional and cognitive recovery in acute stroke patients who were instructed to listen to their favorite music, when compared with audio book listening or a passive control group [30] (for replication and upgrade, see [31]).

Notably, against our expectation, music emotions were not stronger in a familiar listening environment (home) than in a novel environment (laboratory). Although psychological and cortisol-indexed music emotions were modulated by music in both listening environments, they were not stronger at home than in the laboratory. Yet, the effect of the listening environment was seen in the general cortisol levels which were lower at home than in the laboratory. Our expectation was based on an assumption that emotionally safe and familiar environment would support the emergence of stronger emotional responses [22,23,26]. However, the low arousal emotion ratings were always higher in the laboratory than at home. This suggests that even if there was no general effect of the listening environment in music emotional processes, there are other more distinct patterns of the modulation of emotional processes in which the listening environment might be an important factor. Admittedly, since the randomization between the order of listening environments was not feasible, these notions are speculative and might also be contaminated by the fixed order of the environment or other systematic differences between the research environments (see “Limitation of the study” below).

In previous studies of the field, various biomarkers have been used in the context of music listening as evidenced by a recent systematic review ([32]). It was reported that 13 out of 33 biomarkers tested in clinical or non-clinical contexts changed as a response to listening to music. Cortisol was one of the most used biomarkers in these studies, with about half of clinical studies demonstrating a stress-reducing effect of music listening in the cortisol levels, e.g., in pain patients or before/during surgery. Here we observed effects of music listening in physiological (cortisol) and psychological indices in a non-clinical sample and context following only a brief 10-min listening. To our knowledge, this is the first evidence of this kind obtained in a healthy population without induced stress by brief empirical manipulation (for instance by the Trier social stress test, see [19,33]), and also without chronic stress or other clinical conditions.

Limitations of the current study, specified below, are mainly caused by our intention to minimize the burden to the study participants and by the compromises by the participants themselves regarding the completion of the study. Thus, even if a study like ours takes an important step toward ecological validity of the phenomenon under interest, that is, music emotions, there are several issues to be considered as disadvantages of this novel empirical approach.

First, despite the existence of valid questionnaires, we unfortunately have no information about the participants’ self-reported stress level or their personality. In this context it is noteworthy that even in the current study protocol, we were not able to obtain cortisol samples related to music listening at home from 9 out of 37 participants. Having more questionnaires might have decreased their participation even further. Second, although the participants were instructed to conduct the home listening part of the study in the afternoon, corresponding to the timing of the laboratory experiment, not all of them followed this instruction. Third, one problem in cortisol studies is the relatively long time period needed for probing optimally evoked changes in cortisol levels after music listening. In further investigations, the time between the music listening (or any other emotional manipulation) and cortisol measurement should optimally be longer than in the current contribution in which it was 20 min. Fourth, we were not in a position for randomizing the order of the measurements at home vs. in the laboratory. To ensure that the saliva collection and music listening were properly conducted also at home environment by half of the participants before the laboratory environment, we should have had the research assistants to be present at home. However, for financial reasons this was not feasible in terms of their working hours available. Besides this resource issue there is another, even more important psychological issue, namely, the privacy of the participants. Having assistants enter the homes of the participants would have created an additional factor disturbing the listening. Admittedly, the lack of control upon the participants’ music listening, filling in the questionnaires, and saliva sample collection at homes leaves space for speculations upon the origins of the current findings which indicate that the low-arousal emotion ratings were higher in the laboratory than at home while the opposite was true for high-arousal emotion ratings (after listening to neutral music). On the other hand, in the laboratory we conducted ECG and EDA recordings that might have disturbed the listening atmosphere. Furthermore, in the laboratory, research personnel was present, possibly causing some inconvenience even if the listening took place in a recording chamber without any other persons present. However, the baseline emotional ratings of the participants before any music listening in the laboratory vs. at home suggest that the laboratory environment was not experienced as particularly stressful: The high-arousal emotion ratings were lower in the laboratory than at home. The low-arousal emotion ratings were higher in the laboratory than at home. From this we might conclude that the laboratory (including the presence of personnel) was not experienced more stressful than home and could recommend that in future, home-based studies like the current one also have research personnel at home to ensure the participants’ compliance during all stages of the research.

Thus, in general, as exemplified by our study limitations, conducting listening and physiological experiments in two different environments is challenging. Various issues regarding experimental design and practical procedures need to be considered. However, in our view, it is feasible and necessary to upgrade previous laboratory-based investigations by investigations in other environments, also in the fields of music psychology and neurosciences of music.

## 5. Conclusions

In the current contribution, we show that in the context of music listening, the participants’ emotion ratings differed between home and the laboratory. Furthermore, their cortisol levels were generally lower at home than in the laboratory and decreased after music listening at home and in the laboratory. These findings promote the use of music as a tool in professional music therapy, in music medicine, and as “self-medication”. In parallel, they illuminate salivary cortisol as a sensitive index of emotional state even in non-clinical populations and listening environments.

## Figures and Tables

**Figure 1 brainsci-11-00593-f001:**
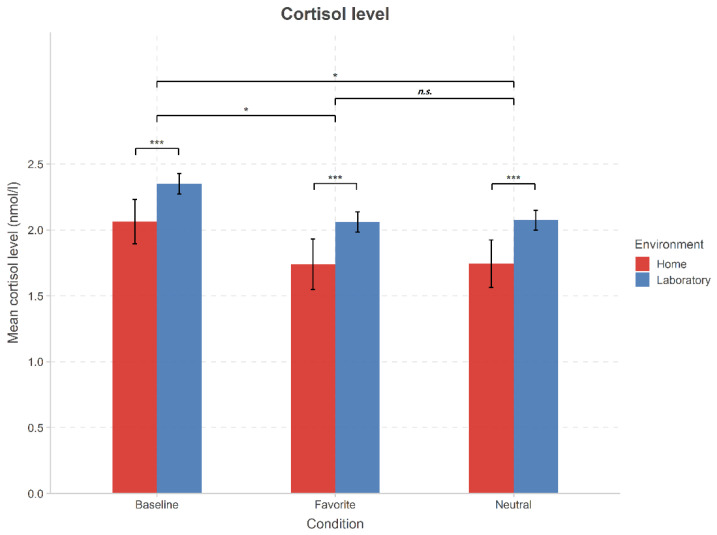
Cortisol levels as collected at home (red) and in the laboratory (blue) (mean and SEM). Cortisol levels were the highest before music listening (baseline) and were significantly decreased after neutral and favorite music listening. Cortisol level were lower at home than in the laboratory. There was no interaction between the listening environment and music listening. * *p* < 0.05, *** *p* < 0.001, n.s. *p* > 0.05.

**Table 1 brainsci-11-00593-t001:** Results of Wilcoxon signed-rank test comparing emotion ratings at home and in the laboratory. Significant results are marked with **bold font** and marginally significant with *italics.*

	Music Type	Median			Z	*p*
		Lab	Home	Lab—Home		
High arousal	Baseline	2.25	2.25	0	−0.776	0.438
	Favorite	2.5	2.375	0.25	−1.494	0.135
	**Neutral**	**2**	**2.25**	**−0.25**	**−2.138**	**0.033**
Low arousal	**Baseline**	**2.5**	**2.25**	**0.25**	**−2.989**	**0.003**
	Favorite	2.75	2.5	0.25	−1.362	0.173
	Neutral	2.25	2.25	0.25	−1.55	0.121
Positive valence	**Baseline**	**3**	**2.5**	**0.375**	**−3.013**	**0.003**
	*Favorite*	*3.25*	*3*	*0.25*	*−1.712*	*0.087*
	Neutral	2.75	2.5	0.125	−1.348	0.178
Negative valence	**Baseline**	**1.5**	**1.75**	**−0.125**	**−2.856**	**0.004**
	*Favorite*	*1.5*	*1.75*	*0*	*−1.881*	*0.06*
	Neutral	1.75	1.75	0	−1.233	0.217

## Data Availability

The datasets collected during the current study are not publicly available due to the fact that the participants were not informed of the possibility that their data would be openly available. In contrast, we promised to keep the data anonymous and out of access of others than the researchers of this project.

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
