# Peer review of "Psychological and Physiological Signatures of Music Listening in Different Listening Environments—An Exploratory Study"

_brainsci, 2021, doi:10.3390/brainsci11050593_

Round 1

Reviewer 1 Report

In this manuscript, the authors investigated the psychological and physiological effects of listening to music in the laboratory vs at home. Little research has examined the influence of listening environment and the current manuscript may provide important preliminary evidence. My main concern with this manuscript are three fold: first, the different procedures employed for laboratory vs home conditions; second, the lack of strict constraint on the experimental conditions at home; and third, the way how the emotional measurement is analyzed.

First, ECG and EDA were measured for the laboratory but not home condition, therefore, there is the possibility that the difference between the two conditions was simply caused by this additional measurement (which generally causes stress).

Second, the authors instructed subjects to conduct all the steps at home on their own, without confirming whether or to what extent the subjects followed the instructions. It is possible that this lack of strict constraint on the study steps itself rather than music listening environment caused the differences observed here.

Third, emotions are generally considered in two dimensions, valence and arousal. The categorization of the emotional ratings into high vs low-arousal may be inappropriate. I suggest that the authors categorize the emotional ratings into four groups based on both arousal and valence (positive vs negative), that will provide more information.

Author Response

See our comments in the attached file.

Reviewer 2 Report

This study presents a relevant topic. The manuscript is easy readable and organized, but still can be benefited after proofreading. My major concern is the revision of the state-of-the-art, as very few studies from the last five years are commented, and compared with the current reported findings. Thus, I cannot appropriately judge the contribution of this study. I have some comments and suggestions, as follows:

  • The should add a section Related Work to comment deeply the state-of-the-art, mainly related studies published between 2016 and 2021. Notice that the current manuscript has a total of 22 references, being 15 (68.18%) published before 2015.
  • The quality of Figures 1 and 2 should be improved.
  • Please provide the heart rate analysis to support  in the section Discussion this phrase "The second hypothesis was confirmed: favorite music did evoke stronger low- and high-arousal emotional ratings than neutral music both at home and in the laboratory. This was corroborated by the heart rate findings which indicated higher heart rates after listening to favorite music than after listening to neutral music." Notice that there are other phrases involving ECG, such as "Here we observed effects of music listening in physiological (cortisol, ECG) and psychological in-dices in a non-clinical sample and context following only a brief 10-minute listening."
  • The authors are not reporting results with Electrocardiogram (ECG) and electrodermal activity (EDA), thus they should avoid this phrase "To our knowledge, this is the first evidence of this kind, obtained with three physiological measures paralleled with emotion ratings, in a healthy population without induced stress by brief empirical manipulation (for instance by the Trier social stress test, see Khalfa et al., 2003), and also without chronic stress or other clinical conditions."
  • Add more recent studies from the last five years to compare and discuss the findings. The current manuscript discusses very few references, almost all published before 2015.

Author Response

See our comments in the attached file.

Reviewer 3 Report

Authors in this paper compared music emotion ratings and their physiological correlates when the participants listened to music at home and in the laboratory. 

My comments to the article are as follows:

- As part of the introduction, I propose to implement a broader background in the field of the impact of music on stress and refer, for example, to the publication: The Impact of Different Sounds on Stress Level in the Context of EEG, Cardiac Measures and Subjective Stress Level: A Pilot Study, Brain Sciences from 2020. Additionally, it will positively influence the updating of the bibliography.

- Please write on the basis of which the participants of the study were selected.

- Why did you choose this and not another analysis of statistical data?

- Figure 1 is not fully legible. Maybe you should consider modifying its appearance.

- The article lacks the Conclusions section - it should be supplemented.

- References to the literature should be made on the basis of square brackets with numbering.

- The article is not formatted according to the generally accepted Brain Sciences format. This should be corrected if the article is finally published.

Author Response

See our comments in the attached file.

Round 2

Reviewer 1 Report

Thank the authors for addressing my first concern. The second concern should be mentioned more clearly in the manuscript as a limitation of the study. And the authors' response to my third concern is disappointing, reanalyzing emotions in their forthcoming paper does not justify that this manuscript can be accepted by an international journal. From the perspective of psychology of emotion or psychiatry, it is common sense to define emotions in terms of valence and arousal, and therefore the current treatment of the emotional scales (i.e., ignoring the valence of emotion) is too preliminary and inappropriate. They should at least present the valence-based analysis in the supplementary material.

Reviewer 2 Report

The revised manuscript is well written and easy readable. I would like to thank the authors for attending my concerns. I have still some minor comments on the Conclusion, as follows:

1- The results about heart analysis from ECG signals are not reported. Then, the authors cannot support these phrases (see section Conclusion) "Furthermore, music listening modulated the heart rate of the listeners in the laboratory recordings.", "In parallel, they illuminate salivary cortisol and heart rate 356 as sensitive indices of emotional state even in non-clinical populations and listening environments."

Author Response

Reviewer 1

The results about heart analysis from ECG signals are not reported. Then, the authors cannot support these phrases (see section Conclusion) "Furthermore, music listening modulated the heart rate of the listeners in the laboratory recordings.", "In parallel, they illuminate salivary cortisol and heart rate as sensitive indices of emotional state even in non-clinical populations and listening environments."

  • Apologies for ignoring these when preparing the R1 version of the manuscript. These have now been omitted.

Reviewer 3 Report

Dear Authors, 

Thank you for the answers provided. However, I have a few more comments.

- As part of the citations, I still propose to consider the reference to the study of the impact of sound on stress using, among others, EEG for example from: The Impact of Different Sounds on Stress Level in the Context of EEG, Cardiac Measures and Subjective Stress Level: A Pilot Study, Brain Sciences from 2020.

- The selection of the group of participants should be described in more detail. Were they random people?

- The answer in terms of statistical analysis is too general. This should be scientifically substantiated. Make a comparison.

Author Response

Thank you for the answers provided. However, I have a few more comments.

- As part of the citations, I still propose to consider the reference to the study of the impact of sound on stress using, among others, EEG for example from: The Impact of Different Sounds on Stress Level in the Context of EEG, Cardiac Measures and Subjective Stress Level: A Pilot Study, Brain Sciences from 2020.

  • This paper is now cited as ref. 33.

- The selection of the group of participants should be described in more detail. Were they random people?

* As reported in R1 revision, these participants were volunteers and thus could be called a convenience sample – not randomized from a larger pool of students, members of a societal organization, or so. This information is now added in the R2 revision (line 105): They were selected on the basis of their availability and interest on participation, the participants thus forming a convenience sample.

- The answer in terms of statistical analysis is too general. This should be scientifically substantiated. Make a comparison.

  • In our analysis pipeline, we first checked whether the data are normally distributed. Regarding cortisol, this was the case after logarithmic transformation and we focused on parametric statistical methods. Then we decided to use ANOVA (with appropriate post hoc tests). More advanced methods (such as linear mixed models) would not, in our view, be necessary or justified since the current data does not provide longitudinal time series data or data from two or more groups of participants (for the sake of comparison, see Linnavalli et al. 2018, https://doi.org/10.1038/s41598-018-27126-5 or Putkinen et al. 2019, https://doi.org/10.1038/s41598-019-47467-z, both from our research group).
  • When re-checking the distributions for emotion ranking and when also considering the comments of Reviewer 1, we decided to re-analyze a subset of the behavioral data by using non-parametric Wilcoxon signed-rank tests. In this work, we included arousal and valence models. For results, see Table 1 and Results, section 2.1.